# Heterotrophy Compared to Photoautotrophy for Growth Characteristics and Pigment Compositions in Batch Cultures of Four Green Microalgae

**DOI:** 10.3390/plants13091182

**Published:** 2024-04-24

**Authors:** Thanh Tung Le, Amélie Corato, Thomas Gerards, Stéphanie Gérin, Claire Remacle, Fabrice Franck

**Affiliations:** 1Laboratory of Bioenergetics, InBios/PhytoSystems, Department of Life Sciences, University of Liège, Chemin de la Vallée 4, 4000 Liège, Belgium; tungrimf@gmail.com (T.T.L.); amcorato@gmail.com (A.C.); thomas.gerards@spw.wallonie.be (T.G.); sgerin@uliege.be (S.G.); 2Research Institute for Marine Fisheries, 224 Le Lai Street, Ngo Quyen District, Hai Phong City 04000, Vietnam; 3Diagenode, Liège Science Park, Rue du Bois Saint-Jean 3, 4102 Liège, Belgium; 4Département de la Recherche et du Développement Technologique, SPW, Place de la Wallonie 1 (B3), 5100 Namur, Belgium; 5Genetics and Physiology of Microalgae, InBios/PhytoSystems, Department of Life Sciences, University of Liège, Chemin de la Vallée 4, 4000 Liège, Belgium; c.remacle@uliege.be

**Keywords:** microalgae, photoautotrophy, heterotrophy, growth, pigment content, photoacclimation, weak light effect

## Abstract

Four strains of green microalgae (*Scenedesmus acutus*, *Scenedesmus vacuolatus*, *Chlorella sorokiniana*, and *Chlamydomonas reinhardtii*) were compared to determine growth and pigment composition under photoautotrophic or heterotrophic conditions. Batch growth experiments were performed in multicultivators with online monitoring of optical density. For photoautotrophic growth, light-limited (CO_2_-sufficient) growth was analyzed under different light intensities during the exponential and deceleration growth phases. The specific growth rate, measured during the exponential phase, and the maximal biomass productivity, measured during the deceleration phase, were not related to each other when different light intensities and different species were considered. This indicates species-dependent photoacclimation effects during cultivation time, which was confirmed by light-dependent changes in pigment content and composition when exponential and deceleration phases were compared. Except for *C. reinhardtii*, which does not grow on glucose, heterotrophic growth was promoted to similar extents by acetate and by glucose; however, these two substrates led to different pigment compositions. Weak light increased the pigment content during heterotrophy in the four species but was efficient in promoting growth only in *S. acutus*. *C. sorokiniana*, and *S. vacuolatus* exhibited the best potential for heterotrophic biomass productivities, both on glucose and acetate, with carotenoid (lutein) content being the highest in the former.

## 1. Introduction

Microalgae are the subject of increasing scientific and economical interest as photosynthetic microorganisms with large potential as sources of economically important products, ranging from biofuels to pharmaceutical drugs. Some species (such as *Chlorella* sp., *Haematococcus pluvialis* or *Dunaliella salina*) are already cultivated on large scales, mainly in photoautotrophic mode. Due to the production cost of microalgal biomass, the microalgal biorefinery concept has emerged in order to combine the exploitation of classes of compounds contained in different biomass fractions [1,2]. Among them, pigments, especially carotenoids, are valuable antioxidants for which relevant species and cultivation processes have been studied [3,4,5]. For carotenoids, a distinction is made between primary carotenoids, which are constitutive components of the photosynthetic apparatus, and secondary carotenoids, which accumulate under stress [5]. In green microalgae, lutein is generally the most abundant primary carotenoid [4], whereas astaxanthin or β–carotene accumulate to high levels under stress in some species. Simple one-stage processes have been developed for photoautotrophic lutein production [4], whereas more complex two-stage processes are usually necessary for controlled secondary carotenoid production of higher value, such as astaxanthin [6].

A recent trend in microalgal research is the exploration of the potential of heterotrophic growth for less costly biomass generation and its use for fatty acid and pigment production, among others [7,8,9,10]. There are two main advantages of heterotrophy over photoautotrophy: indoor fermenters with low land requirements can easily be scaled up, and higher cell densities can be obtained, since no limitation of light availability occurs as the density rises. The most commonly used organic carbon substrate is glucose, which can be supplied at a relatively high concentration (typically 10–30 g·L^−1^) in several *Chlorella* species (reviewed in [9]). Glycerol and acetate have also been investigated, but in the case of the latter, toxicity at high concentrations was reported for green microalgae, due to the pH rise with acetate consumption and to Na^+^ ion accumulation [8]. As far as pigment production is concerned, heterotrophic processes do not seem the best choice at first glance, because the photosynthetic apparatus is not involved in growth; therefore, pigment biogenesis is likely to be slowed down. In green microalgae, the chlorophyll and carotenoid content of the biomass are generally lower in darkness than in light [11], an effect which was ascribed to photoreceptor (phototropin) control over gene expression for *Chlamydomonas reinhardtii* [12]. If pigment biosynthesis is controlled by photoreceptors, one may expect that weak, non-photosynthetic light may enhance the pigment content of the biomass in the heterotrophic mode. Additionally, weak light may also enhance heterotrophic growth, as reported for *Chlorella vulgaris* on glucose [13]. On the other hand, pigment biosynthesis in the dark seems much more dependent on species, and for some species of green microalga (e.g., *Chlorella protothecoides*) heterotrophic production of lutein is considered for optimization [14,15]. To our knowledge, the effect of the carbon source on pigment biosynthesis in the dark has not been thoroughly studied.

In this work, we aimed to evaluate the potential of heterotrophy for growth and pigment accumulation in green microalgae in comparison to light-limited (CO_2_-sufficient) photoautotrophy. However, the comparison of growth performances between these two conditions is made difficult for the following reasons. First, results obtained in photoautotrophic conditions will be dependent on the particular geometry of the growth device (mainly its thickness). They will also depend on the incident light intensity used, since growth is driven solely by photosynthesis. In addition, photoacclimation leading to adjustments in pigment content may occur during batch growth due to progressive shading [16,17]. A range of growth rates and of pigment compositions must, therefore, be obtained after proper definition of growth phases (exponential and deceleration phases), obtained under a range of incident light intensities. For heterotrophic conditions, no such difficulties are expected, and growth is essentially determined by the availability of the carbon source and sufficient aeration [18], as long as no autoinhibiting substance accumulates.

In order to compare heterotrophy and photoautotrophy, we chose *Chlamydomonas reinhardtii*, *Chlorella sorokiniana*, *Scenedesmus acutus*, and *Scenedesmus vacuolatus* (also known as *Chlorella emersonii*) as representatives of species that are commonly used for research and/or large scale applications. In order to ensure a sufficiently short light path for photoautotrophic growth, cultures were performed in multicultivators equipped with glass tubes of small diameter (3 cm) illuminated from one side. Photoautotrophic growth and pigment composition were analyzed as a function of light intensity ranging from 100 to 700 µmol·m^−2^·s^−1^ and were then compared to results obtained for heterotrophy. Additional objectives in this work were to compare the suitability of acetate and glucose as substrates for heterotrophic growth and pigment accumulation, and to evaluate the possibility of stimulating heterotrophic pigment accumulation using weak light.

## 2. Results

### 2.1. Comparison of Growth Characteristics in Light-Limited Photoautotrophy and in Heterotrophy

#### 2.1.1. Dependence of Growth on Light Intensity in Light-Limited Photoautotrophy

Four species of green microalgae (*Chlamydomonas reinhardtii*, *Chlorella sorokiniana*, *Scenedesmus acutus*, and *Scenedesmus vacuolatus*) were compared for their photoautotrophic growth performances in multicultivators under bubbling with CO_2_-supplemented air. As observed in several studies, CO_2_ supplementation is necessary to reach avoid CO_2_ limitation of growth in dense photoautotrophic cultures [19,20,21,22]. For this purpose, batch growth experiments were conducted over 72 h, and the optical density (OD) signal from the multicultivator was processed as indicated in the Material and Methods (Section 4.3.2). Examples of OD growth curves for the four species at an incident light intensity of 300 µmol m^−2^ s^−1^ are shown in Figure 1. The biomass density values were measured at the end of the 72 h cultures. The OD growth curves typically show a short exponential phase (15 to 30 h duration depending on species and light intensity) followed by a long deceleration phase, often called the linear phase in previous works [23].

For photoautotrophic cultures, the exponential phase, during which the cell division rate is constant, is usually characterized by the specific growth rate (µ in h^−1^), which is constant during this phase. The maximum rate at which biomass accumulates is observed during the first hours of the deceleration phase (often called the linear phase) that follows. This maximum biomass accumulation rate is most usefully expressed as maximum volumetric biomass productivity (V_max_ in g L^−1^ day^−1^) [23,24]. In the following, we used OD growth curves to estimate these two parameters for light-limited photoautotrophic cultures (see Material and Methods, Section 4.3.2). The effect of light intensity in the range 100–700 µmol m^−2^ s^−1^ on µ (exponential phase) and V_max_ (deceleration phase) is shown in Figure 2 for the four species.

Except for *C. reinhardtii*, µ tended to show a maximal value (µ_max_) in the light intensity range tested (around 500 µmol m^−2^ s^−1^ for *C. sorokiniana*, 300 µmol m^−2^ s^−1^ for *S. acutus*, and 500 µmol m^−2^ s^−1^ for *S. vacuolatus*). The highest µ values (around 0.15 h^−1^, corresponding to a generation time of 4.6 h) were found for *C. reinhardtii* at 700 µmol m^−2^ s^−1^, and for *C. sorokiniana* at 500 µmol m^−2^ s^−1^.

In the four investigated species, the maximum biomass productivity (V_max_) found at high biomass density increased with light intensity. In contrast to µ_max_, V_max_ did not clearly reach a maximum within the light intensity range tested, except for *S. vacuolatus*, where growth saturation was visible at 700 µmol m^−2^ s^−1^. The different light-dependance of V_max_ and µ_max_ was particularly evident for *S. acutus*, for which a linear increase in V_max_ was observed. This species also showed the highest V_max_ value (around 1.8 g L^−1^ day^−1^). In a general way, these results indicate the absence of a relationship between the two growth parameters, µ (exponential phase) and V_max_ (deceleration phase), when different species are compared.

#### 2.1.2. Heterotrophic Growth: Effect of Organic Carbon Source (Acetate, Glucose) and of Weak Light

The four species could grow heterotrophically in complete darkness on glucose or acetate as organic carbon sources, except *C. reinhardtii*, which is well-known to use only acetate for heterotrophic growth [25]. In this study, glucose was used at a concentration of 15 g L^−1^ (absence of growth inhibition at this concentration was checked in preliminary experiments, data not shown) and acetate was used at a concentration of 2 g L^−1^, which is usual for *C. reinhardtii* [25]). Multicultivator OD growth curves were processed using the same procedure as for photoautotrophic experiments (Material and Methods, Section 4.3.2). Only µ_max_ was considered to characterize the heterotrophic growth rate, since no limitation to growth (like light-limitation due to shading during photoautotrophic growth) is observed before exhaustion of the organic carbon source, provided the supply of oxygen is sufficient.

Values of µ_max_ for the four species on acetate or glucose were compared. Furthermore, possible stimulation of growth by weak light was investigated. Weak light is defined here as a light of intensity low enough not to sustain photoautotrophic growth. With the multicultivator device used in this study, we found that light of 5 µmol m^−2^ s^−1^ provided by the white LEDs of this apparatus met this criterium.

As shown in Figure 3, two species (*C. sorokiniana* and *S. vacuolatus*) showed fast growth rates both on glucose and acetate, whereas the other two species (*S. acutus* and *C. reinhardtii*) showed much slower growth (representative growth curves are shown in Appendix A for the four species). The two ‘fast’ species showed µ_max_ values (between 0.08 and 0.13 h^−1^) that were of the same order as those found in photoautotrophic conditions in the absence of CO_2_ limitation (see Figure 2). In contrast, the µ_max_ values of the two ‘slow’ species were much lower (between 0.020 and 0.035 h^−1^). It is noteworthy that for the three species that could accept glucose as carbon substrate, there were no marked differences between growth rate on glucose or acetate (the most important difference, of 33%, was observed with *C. sorokiniana*, which showed some preference for glucose).

A previous study reported that the growth of *Chlorella vulgaris* on glucose was stimulated by light fluxes weak enough not to support net photoautotrophic growth [15]. Here, weak light did not seem to increase µ_max_ values in the four studied species (Figure 3). However, the somewhat higher µ_max_ value found with *S. acutus* with weak light compared to the dark control on glucose (Figure 3) attracted our attention, which led us to perform growth experiments on longer time scales than those needed for µ_max_ evaluation. Average growth curves recorded over six days are shown for this species in Figure 4 in three conditions: weak light only, glucose in darkness, and glucose under weak light. The control without glucose indicates the absence of growth under the weak light intensity (5 µmol m^−2^ s^−1^), meaning that there is no net photosynthesis in this condition. When comparing the two growth curves obtained with glucose, with or without weak light, it appears that growth was similar during the first 50 h, but that it later became significantly slower in complete darkness. After six days, optical density values were more than doubled with weak light. Semi-logarithmic plots indicate that exponential growth was better maintained with weak light than without it. We did not observe similar weak light effects with the other three species.

The biomass yield on carbon substrate (Y_X/S_) is an important parameter of heterotrophic microalgal cultures. Estimations of the biomass yield on glucose are consistently around 0.5 g(DW) g^−1^ in many studies on various green microalgae [7], a value similar to those also found in this study (Table 1). Reported biomass yields on acetate are more variable [7]. Here, we also estimated the biomass yield on acetate for three species and we obtained a consistent value of around 0.3 g(DW) g^−1^ (Table 1). These calculations were made on the assumption that all glucose or acetate was consumed when growth stopped, and by dividing the obtained biomass densities by the initial glucose or acetate concentrations.

The rather low values for the biomass yields on acetate obtained this way prompted us to verify acetate consumption by HPLC (Appendix A). With *S. vacuolatus* grown heterotrophically, the biomass yield on acetate (Y_X/acetate_) calculated on the basis of acetate consumption was 0.30 ± 0.03 g(DW) g^−1^, close to the values indicated in Table 1. Acetate was completely consumed when growth stopped, although the pH rose to a value of 8.7 at the stationary phase (Appendix A).

### 2.2. Comparison of Pigment Content of the Biomass in Light-Limited Photoautotrophy and in Heterotrophy

The four strains were compared for the content of their biomass in chlorophylls and carotenoids, when grown in light-limited photoautotrophy and in heterotrophy on glucose or acetate.

#### 2.2.1. Effect of Light-Intensity on Pigment Content during Exponential and Deceleration Phases of Light-Limited Photoautotrophic Growth

It is well-known that microalgae are endowed with photoacclimation capabilities which lead to changes in pigment content as a function of perceived light quantity. In many species, a reduction in pigment content has been reported as a photoacclimation response to higher light intensity [16,17,26]. Such changes can occur within a few generations [27], so for fast growing strains, it is expected that pigment content can change during batch cultivation. During the exponential phase, shading is not pronounced; therefore, cells will be photoacclimated to a light intensity value close to that of incident light. As the cell density increases with time, the average light intensity at the level of each cell will strongly decrease, and photoacclimation to lower light is expected. Any comparison between heterotrophic and photoautotrophic conditions should consider these photoacclimation responses. Therefore, in order to characterize the pigmentation of photoautotrophically grown cells, we analyzed the pigment content of their biomass under different light intensities (from 100 to 700 µmol m^−2^ s^−1^) in two well-defined growth stages: during the exponential phase (20–25 h) and at the late deceleration phase (70–80 h).

Figure 5 shows chlorophylls *a* + *b*, lutein, and carotene content of the biomass of the four species at the two phases when grown at different light intensities. For the chlorophylls, it is expected that their content will decrease with light intensity as a result of photoacclimation during exponential phase. This behavior was indeed observed clearly for *S. acutus* and for *S. vauolatus*, with these two species also showing a lower chlorophyll content among the four strains no matter the light intensity (Figure 5A). *C. sorokiniana* showed a slightly decreasing trend for light intensities higher than 300 µmol m^−2^ s^−1^. *C. reinhardtii* showed an atypical behavior, with a slight increase in chlorophyll content from 100 to 300 µmol m^−2^ s^−1^ and no significant variation for higher intensities.

The highest chlorophyll content values, around 5% of DW, were found in *C. sorokiniana* and in *C. reinhardtii*. As a result of photoacclimation to lower perceived light, the chlorophyll content was generally higher during the deceleration phase compared to the exponential phase (Figure 5B), except for *C. reinhardtii*, in which it varied only slightly. Even though strong shading occurred, the effect of light intensity on chlorophyll content was still visible during the deceleration phase. This effect was the strongest for *S. acutus*, which showed around twice higher chlorophyll content at 100 µmol m^−2^ s^−1^ compared to 700 µmol·m^−2^·s^−1^, and it was the weaker for *C. reinhardtii*, which showed only weak changes.

The carotene content (Figure 5E,F) responded to light intensity and to growth phase in a manner similar to that of the chlorophyll content. Again, *C. reinhardtii*, which had the highest carotene content of the four strains, showed only minor responses to light intensity.

Lutein, the most abundant xanthophyll in green microalgae, showed a significantly different pattern (Figure 5C,D). It was not found to decrease in content as a response to increasing light intensity in any of the four species (a slight effect of this kind was observed only for *S. acutus* during deceleration phase). In several cases, the content of lutein showed an increase from 100 to 300 µmol m^−2^ s^−1^ and no or only minor variation for stronger light intensities. This trend was significant during the exponential phase for *S. acutus*, *C. sorokiniana*, and *C. reinhardtii*. It was also found during the deceleration phase for *C. sorokiniana*, *C. reinhardtii*, and *S. vacuolatus*. When comparing lutein content values in the deceleration versus exponential phases, it appeared that it was much increased during the deceleration phase for *C. sorokiniana* and for *S. vacuolatus*, whereas no such large phase effect was observed for the two other species. Of the four investigated species, our *C. sorokiniana* strain was the richest one in lutein, which reached up to 0.6% of DW during the deceleration phase, a value similar to that found in a recent study on another strain of the same species [28].

#### 2.2.2. Pigment Content of Heterotrophically Grown Cells: Effect of Carbon Source (Glucose or Acetate) and of Weak Light

Chlorophylls *a* + *b*, lutein, and carotene were quantified during the exponential growth phase which, in heterotrophy, can be maintained for longer times than in photoautotrophy. For valid comparisons, we choose to collect samples after 3–4 generations from the start of the culture in order to minimize the effect of the differences in growth rate among the four species (Figure 6).

In total darkness, the chlorophyll content was highest in *C. reinhardtii* and in *C. sorokiniana* when acetate was used, and it was lowest in *S. vacuolatus* whatever the carbon source (Figure 6A). For the species which could grow either on glucose or on acetate, the carbon source had a marked effect on the chlorophyll content: for *S. vacuolatus* and for *C. sorokiniana*, acetate promoted chlorophyll content better than glucose, whereas no significant difference was found for *S. acutus*. This effect of the carbon source was also observed for the lutein and carotene content values in the same species-dependent manner (Figure 6B,C). Weak light caused a general increase in pigment content whatever the species or carbon source. This effect was particularly evident for *S. acutus*, with a general doubling of pigment content promoted by weak light (Figure 6A–C). The pigment content in this species was found similar to that observed under photoautotrophy at the weakest growth light. For *C. sorokiniana* and *S. vacuolatus*, however, weak light could not promote pigment accumulation to levels as high as those found under photoautotrophy. For the richest pigment producer in photoautotrophy, *C. sorokiniana*, the chlorophyll, lutein, and carotene content values were, respectively, 29%, 25%, and 23% of their maximum when this species was grown heterotrophically on acetate with weak light. *C. reinhardtii* had unique features, in that it maintained high levels of carotene under heterotrophy (around 0.2% with weak light, similar to photoautotrophy) while other pigments (chlorophylls, lutein) decreased when compared to their maximum values under photoautotrophy.

## 3. Discussion

### 3.1. Comparing Photoautotrophic and Heterotrophic Growth Performances on the Basis of Batch Experiments

This study confirms previous ones showing that light-limited photoautotrophy involves a rather short exponential phase followed by a long deceleration phase (often called the linear phase) [23,29,30,31]. This is observed here for the four investigated species of green microalga. Based on OD growth curves, the most useful growth parameter is the maximum biomass productivity (V_max_, obtained during deceleration phase), because it gives productivity in conditions of optimal biomass concentrations that should also apply to large-scale photoautotrophic cultures in a semi-continuous mode. It is shown here clearly that V_max_ has no evident relationship with the specific growth rate (µ), a commonly used parameter to define growth during exponential phase. An absence of correlation between these two parameters was already pointed out by Ogbona et al. [23] for *Chlorella pyrenoidosa* and *Arthrospira platensis* when comparing different types of photobioreactors, and it was concluded that the linear growth rate (equivalent to V_max_ here) was a more useful growth index than the specific growth rate (µ).

The absence of correlation between µ and Vmax can be explained by considering that µ is measured at low cell densities and indicates the pace at which cells divide when fully exposed to a given light intensity. Its dependence on light intensity reflects the photosynthetic light curve, with saturation at relatively low light intensity (generally around 400 µmol m^−2^ s^−1^ PAR [32,33]). In contrast, V_max_ is measured at high cell densities, and is expected to show a more linear relationship over a broader range of light intensities [34,35] because shading decreases the average light intensity inside the culture, which leads in turn to a higher average photosynthetic efficiency. In this situation, photosaturation will only occur in the most exposed culture region. Species-dependent variations in maximal productivity at high cell densities may be related to different abilities to reach a compromise between the need to limit the negative impact of photosaturation in the most exposed regions and to maintain high photosynthetic efficiency in shaded regions, by adjusting pigment content and maintenance energy losses.

For heterotrophic cultures µ_max_ is the most adequate parameter to define growth performances, because it is expected that exponential growth can be maintained almost until the carbon source is exhausted. Only indicative V_max_ values can be given on the basis of batch cultures, for which the initial carbon substrate is progressively depleted, so that V_max_ will be reached close to the cessation of growth. It is likely that biomass productivity can be increased beyond this value in fed-batch cultures by continuous or semi-continuous feeding with a carbon substrate, as shown previously for some *Chlorella* species [36,37]. On the basis of the present study, it is possible to show that for the two strains with fast heterotrophic growth, *S. vacuolatus* and *C. sorokiniana*, V_max_ is higher in heterotrophic conditions on glucose than in light-limited photoautotrophic conditions at the highest light intensity (700 µmol m^−2^ s^−1^). This is shown in Figure 7, where biomass density growth curves were established in these two conditions on the basis of OD growth curves and of the relationship between OD and biomass density. With these two species, short batch growth experiments demonstrate the advantage of glucose-driven heterotrophy over photoautotrophy. Although µ_max_ values were not higher in heterotrophy than in photoautotrophy, V_max_ values were strongly increased due to the extended length of the exponential phase. Biomass productivities around 5 g L^−1^day^−1^, such as found here, are in the highest range of published values [38]. For ‘slower’ species, and for acetate-driven heterotrophy, it would be possible to extrapolate growth on the basis of µ_max_ values in order to determine how long it would take for the biomass productivity to surpass the photoautotrophic V_max_.

A striking result of this study is also that the heterotrophic growth rate of *S. vacuolatus*, *C. sorokiniana*, or *S. acutus* seems only weakly dependent on the nature of carbon substrate (glucose or acetate), as revealed by µ_max_ values. However, large species-dependent variations in µ_max_ are observed (Figure 3). This indicates that heterotrophic growth is rather limited by the general metabolic capacity in the absence of light, rather than by uptake capacity of one or the other substrate or by induction of enzymes specific to the assimilation of the carbon source (such as glyoxylate cycle enzymes for acetate) [8].

In this work, we found that the biomass yield on carbon substrate (Y_X/S_) is around 0.5 for glucose, similar to values found in other studies [7], but that it is only around 0.3 for acetate. In previous works, Y_X/acetate_ values of 0.48 and 0.55 were reported for *Chlorella regularis* [39] and *C. reinhardtii* [36], respectively. However, in their pioneering study on the heterotrophic growth of *Chlorella pyrenoidosa*, Samejima and Myers [40] reported a much lower value of 0.10. It is possible that variations in Y_X/acetate_ values are related to the use of different nitrogen sources due to different redox states of N (nitrate in this study and in [40], urea in [36,39]). This point would require further investigation.

### 3.2. Photoacclimation Processes Revealed by Changes in Pigment Content during Light-Limited Photoautotrophic Batch Growth

It has been observed since early studies on microalgae that the cellular chlorophyll content is subjected to changes in response to growth light intensity, with a general trend of an inverse relationship between chlorophyll content and light intensity. In CO_2_-unrestricted batch cultures, such as here, cells are exposed to the full incident light during the exponential phase, whereas pronounced light attenuation develops during the deceleration phase. Our pigment analyses during the exponential phase and during the deceleration phase indicate that photoacclimation occurs during both phases, but with an extent that is much species-dependent. Obviously, the extent of changes in pigment content during batch growth are dependent on kinetic aspects, in such a way that species capable of fast changes will undergo visible photoacclimation in the time lapse of the batch experiment. In this regard, *C. reinhardtii* and *S. vacuolatus* are two extreme cases found in this study. *C. reinhardtii* showed only weak changes in pigment content in response to light intensity and to light attenuation. *S. vacuolatus* showed marked light intensity effects on pigment content during the exponential phase, as well as a marked increase in pigment content in deceleration phase (Figure 6). Interestingly, increasing incident light intensity entailed decreases in both chlorophyll and carotene content during the deceleration phase in this species, as well as in *C. sorokiniana* and in *S. acutus*, eventhough light attenuation was more effective at higher incident light intensities due to faster growth. This suggests that flashing light perception (due to cell movement across the light path) influences photoacclimation to some extent. In a general way, photoacclimation at high cell densities is an important aspect of microalgal biotechnology. It was shown that high pigment content due to low light acclimation at high cell density has a negative impact on the overall light-use efficiency of a culture, so that one expects increased efficiency in low-chlorophyll mutants [32,33,41].

Our study reveals that carotene and lutein behave quite differently during photoacclimation, with the former following the same trends as chlorophyll, and the latter showing a more complex response. Among the four species, *C. sorokiniana* was the richest lutein source, and the content of this pigment tended to increase with light intensity up to 300 µmol m^−2^ s^−1^, but the content was also higher during the deceleration phase compared to in the exponential phase (Figure 6). A similar pattern was found for *S. vacuolatus*. This unexpected response may be related to the dual function of lutein. This pigment is part of photosynthetic pigment–protein complexes (LHC’s), where it contributes to light-harvesting, and it is also involved in chloroplast protection against reactive oxygen species [42].

In a general way, this part of our study has shown that pigment change analysis during light-limited photoautotrophic batch growth can easily reveal the photoacclimatory capabilities of different algal species. The fact that *C. reinhardtii* only showed minor pigment changes was somewhat unexpected, because previous studies have reported photoacclimatory responses of pigment content in this species [43]. The rather ‘inert’ behavior here may be due to temperature (30 °C rather than 20–25 °C in most studies), strain-specific traits, or slow photoacclimation kinetics in this species.

### 3.3. Species- and Substrate-Dependent Pigmentation during Heterotrophy in Darkness or under Weak Light

Microalgae often contain fewer pigments in darkness than in light [11,12]. One of the objectives of this study was to determine how this difference depends on species and on the particular pigment considered. Here, we found that pigment content was better maintained on acetate than on glucose, except for with *S. acutus*. This species was also the slowest strain for growth in heterotrophy. Substrate-specific effects on pigment accumulation in heterotrophy have not been investigated thoroughly yet, but it is known that glucose shows an inhibitory effect on chlorophyll biosynthesis [44].

For comparisons, in Table 2, we show the pigment content values expressed in percentages of their content during the deceleration phase of photoautotrophic growth in high light (average values between 100 and 700 µmol m^−2^ s^−1^ taken as references). Weak light effects on pigment content values are also indicated.

Table 2 highlights the evidence that, among the four studied species, the two fast-growing strains (*C. sorokiniana* and *S. vacuolatus*) had very weak pigment content values under heterotrophy compared to photoautotrophy. The two ‘slow’ strains (*S. acutus* and *C. reinhardtii*) (see Figure 4) maintained higher pigment content values. This was not an effect of the number of generations from the start of the cultures, since care was taken to collect samples after 3–4 generations. Species-dependent pigment content variations under heterotrophy compared to photoautotrophy were noted also by Sutherland and Ralph [10] for four Scenedesmaceae species at stationary phases and without CO_2_ supplementation during photoautotrophic growth.

It appears for the four species studied here that exposure to weak light (5 µmol m^−2^ s^−1^) during heterotrophic growth led to higher content values in both chlorophylls and carotenoids. This weak light effect was also more obvious for the two slow strains (Table 2). In order to better characterize this effect, it will be necessary to precisely assess the effects of light quality and quantity. In previous studies, it was shown that, in *C. reinhardtii*, the phototropin photoreceptor controls the expression of genes involved in chlorophyll biosynthesis and of the genes encoding chlorophyll-binding proteins [12]. Light was also found to exert control on the expression of genes involved in carotenoid biosynthesis in *C. reinhardtii* [45]. In addition to control at the genetic level, light is important at the biosynthetic step of protochlorophyllide reduction during chlorophyll biosynthesis. Photosynthetic microorganisms possess a light-dependent protochlorophyllide reductase, besides a dark active one [46,47]. In total darkness, only the light-independent pathway is active and, therefore, less chlorophyll accumulation may be expected. From a biotechnological perspective, weak light effects could be exploited to enhance pigment productivity in large-scale heterotrophic cultivation processes.

In contrast to the apparently ubiquitous effect of weak light on pigment content, the same weak light did not accelerate heterotrophic growth in significant ways, except in the case of *S. acutus* grown on glucose, for which this effect was very clear (Figure 5). Reports on weak light effects on growth are scarce in the literature. An effect of this kind was first reported in Killam and Myers [15] for a *Chlorella* strain, but this was later contradicted [48]. Dim light stimulation of cyanobacterial growth on glucose was also reported [49]. In more recent works, heterotrophic cultivation of microalgae in bioreactors are sometimes performed with a supply of weak light [50]. The biological basis of such an effect in terms of light signaling has not been explored at present. We conclude that a supply of weak light can be used as an efficient way of increasing the productivity of valuable pigments in heterotrophic production processes.

## 4. Materials and Methods

### 4.1. Microalgal Strains and Growth Media

*Scenedesmus vacuolatus* (SAG 211.11n) was obtained from the Culture Collection of Algae (SAG), Germany. This strain is synonymous with *Chlorella emersonii*, *Graesiella emersonii*, or *Chlorella fusca* var. *vacuolata* (CCAP 211/11N). Its assignment to the genus *Scenedesmus* was proposed on the basis of sequence analysis (18S RNA) as well as due to its biochemical traits (synthesis of secondary carotenoids) [51]. *Chlamydomonas reinhardtii* (CC1690) was obtained from the *Chlamydomonas* Resource Center, United States. *Chlorella sorokiniana* and *Scenedesmus acutus* (synonyms: *Scenedesmus obliquus* and *Tetradesmus obliquus*, respectively [52]) were collected in the Liège region (Belgium) and identified genetically by sequencing the 18S rRNA gene. DNA extractions were performed following a modified protocol from Newman et al. [53]. PCR amplifications were carried out with the following primers: 5′-GTAGTCATATGCTTGTCTC-3′ (forward) and 5′-GGCTGCTGGCACCAGACTTGC-3′ (reverse) for *C. sorokiniana* [54]; 5′-CTGTGAAACTGCGAATGGCTC-3′ (forward) and 5′-TTTCCTGCTTGGCCTCTAGC-3′ (reverse) for *Scenedesmus acutus*. Obtained PCR products were sent to Genewiz (Germany) for Sanger sequencing. DNA sequences were analyzed with the Nucleotide Basic Local Alignment Search Tool (BLASTn) of NCBI. For *C. sorokiniana*, 99.20% of identity was found with a query cover of 94% and an E-value of 0.0 (Accession number: KY054944.1). For *S. acutus*, 99.84% of identity was found with a query cover of 99% and an E-value of 0.0 (Accession number: MH307949.1).

Pre-cultures were maintained on Bold’s basal medium [54] with a triple quantity of NaNO_3_ (20 mM) (3N-BBM).

Photoautotrophic cultures of the four strains were performed using TMP (Tris-minimum-phosphate) medium [55] with the following composition per liter: 0.0986 g MgSO_4_·7H_2_O, 0.05 g CaCl_2_, and 0.715 g K_2_HPO_4_; 0.3605 g KH_2_PO_4_, 1.7 g NaNO_3_, 0.12 g MgSO_4_, 2.423 g Tris buffer (C_4_H_11_NO_3_), 0.05 g Na_2_EDTA·2H_2_O, 0.0114 g H_3_BO_3_, 0.022 g ZnSO_4_·7H_2_O, 0.00506 g MnCl_2_·4H_2_O, 0.0049 g FeSO_4_·7H_2_O, 0.00161 g CoCl_2_·6H_2_O, 0.00157 g CuSO_4_·5H_2_O, 0.0011 g (NH_4_)_6_Mo_7_O_24_·4H_2_O, and pH 7.2. Heterotrophic cultures of the four strains were performed using the same medium, supplemented with 15 g glucose (TGP: Tris-glucose-phosphate medium) or 2 g acetate (TAP: Tris-acetate-phosphate [55]). All media were autoclaved at 120 °C for 15 min to ensure axenic conditions before any experiments.

### 4.2. Cultivation Conditions

#### 4.2.1. Pre-Cultivation

Pre-cultures were maintained on 3N-BBM medium in Erlenmeyer flasks on an orbital shaker under illumination with fluorescence light of around 50 µmol m^−2^ s^−1^. Actively growing cells were used as starters for experimental cultures at an optical density (750 nm) around 0.2.

#### 4.2.2. Cultivation Conditions

All experiments were carried out in triplicate at 30 °C in MC 1000-OD-8X multicultivators (Photon Systems Instruments, Drásov, Czech Republic), using 100 mL glass tubes containing 60 mL of algal suspension. Photoautotrophic cultures were bubbled using commercial air supplemented with 5% CO_2_ at a rate of 30 mL min^−1^ and were also exposed to light from the white LEDs of the multicultivators at intensities varying from 100 to 700 µmol m^−2^ s^−1^. Heterotrophic cultures using glucose or acetate were either performed in darkness (by covering the multicultivator with black fabric) or under a low light of 5 µmol m^−2^ s^−1^, provided by the white LEDs of the multicultivator, which was unable to sustain growth in the absence of an organic carbon source. Aeration was obtained by bubbling with natural air. Absence of bacterial contamination was checked using optical microscopy observation. For all cultures, pH was checked at the end of the experiment. Except for heterotrophic cultures on acetate showing important increases in pH (see Appendix A), minor increases in pH were found after 72 h, with pH ranging from 7.5 to 8.2, depending on conditions. Previous works have shown that for green microalgae, pH variations in this range have only small effects on growth rates [56,57].

### 4.3. Sampling and Analysis

#### 4.3.1. Biomass Estimation

Biomass dry weight (DW) was measured from 50 mL or 10 mL culture samples taken during the exponential phase (20–30 h) or during the late deceleration phase (60–70 h), respectively. Samples were centrifuged at 5000× *g* for 10 min and washed twice with distilled water. Pellets were then dried for at least 24 h at 80 °C in an oven, and the resulting dry biomass was weighed.

#### 4.3.2. Specific Growth Rate and Productivity Determination

Multicultivators provide online monitoring of optical density at 720 nm (OD(MC) in the following). However, as noted by Vonshak et al. [58], we found that this apparent optical density is not linearly related to true optical density, as measured in a conventional spectrophotometer, especially at high cell densities. The relationship between OD(MC) and true optical density (OD(750), measured at 750 nm in a lambda 20 UV/Vis spectrophotometer, Perkin-Elmer, Wellesley, MA, USA) was, therefore, investigated for correction purposes. A general equation, as follows, was found suitable to correct the OD(MC) signal:(1)$$\frac{OD(750)}{OD(MC)}=a*e^{b*OD(MC)}$$
where ‘*a*’ and ‘*b*’ are species- and condition-dependent constants. In order to determine these constants for each strain and condition, OD(MC) and OD(750) were measured on different dilutions of dense cultures obtained in the given condition. Then, the relationship between the two series of values was fitted using Equation (1) (Appendix A). ‘*a*’ and ‘*b*’ values giving the higher correlation coefficient were retained for further application of Equation (1) in order to correct OD(MC) curves (see an example in Figure 8, Left panel). The correction procedure was further validated by point measurements of OD during the course of the culture (Figure 8, Left panel, circles). It was checked that, for a given species, differences in ‘*a*’ and ‘*b*’ values were not significant when comparing the exponential and deceleration phases.

The specific growth rate µ (expressed in h^−1^) was calculated during exponential phase on the basis of the semi-logarithmic plot of the corrected optical density (OD(750)), which showed a well-defined linear part located in the time region between 5 h and 20 to 30 h depending on light intensity and species (example in Figure 8, Right panel). For photoautotrophic conditions, the maximum volumetric biomass productivity V_max_ was calculated from the slope of the linear part of the deceleration phase of the corrected OD(750) curve (Figure 8, Left panel), and expressed in g L^−1^ day^−1^ after having established the relationship between OD(750) and biomass density for each species and culture condition. This relationship did not change significantly over time during batch cultivation when comparing exponential and late deceleration phase.

#### 4.3.3. Pigment Analysis

Pigment samples were obtained by centrifuging 2 mL samples in the exponential phase and 1 mL samples in the deceleration phase. Samples were centrifuged at 16,000× *g* for 2 min, and then kept at −80 °C for subsequent analyses. Pigments were extracted in a mixture of methanol and dichloromethane (3:1, *v*/*v*). Cells were broken by glass beads using a Tissue Lyser II disrupter (Qiagen, Antwerp, Belgium) at 25 Hz for 5 min. After that, the samples were shaken in Vibrax (VXR basic, VWR, Leuven, Belgium) with the solvent mixture for 30 to 60 min, depending on species. Samples were then centrifuged at 16,000× *g* for 5 min at 4 °C. The supernatant was collected and filtered through a 0.22 µm PTFE membrane prior to HPLC analyses.

Reverse-phase HPLC analysis was performed on a NovaPak C18 column (3.9 × 150 mm, 4 µm particle size, Waters, Milford, MA, USA) with a Shimadzu apparatus (Shimadzu, Kyoto, Japan) equipped with an injector (SIL-20AC), a pump (LC-20AT), a degassing unit (DGU-20A5R), an oven (CTO-10ASVP), a photodiode array detector (SPD-M20A), and a communication module (CBM-20A). The analyses were performed at 25 °C using a flow rate of 1 mL min^−1^. The elution protocol was modified from [59], with the following solvents: 80% methanol + ammonium acetate 100 mM (A), 90% acetonitrile in water (B), and 100% ethyl acetate (C). The gradient was: 0 min—100% A; 0.5 min—100% B; 1.1 min—90% B + 10% C; 6.1 min—65% B + 35% C; 11.5 min—40% B + 60% C; 15 min—100% C; 17 min—100% A; 23 min—100% A. The data were analyzed with the Empower software (Waters, version 6.1.2154.917), and the quantification was based on the peak area at 430 nm, compared with standard solutions (DHI). Alpha- and beta-carotene peaks largely overlapped; therefore, the two carotenes were computed as total carotene.

#### 4.3.4. Quantification of Acetate by HPLC

The HPLC device (Shimadzu, Kyoto, Japan) was composed of an injector S(IL-20ACXR), a pump (LC-20ADXR), a degassing unit (DGU-20A3R), an oven (CTO-20A), a communication module (CBM-20A), and a refractive index detector (RID-20A). The software LabSolution (Shimadzu) was used to control the equipment and to collect and analyze the data. The separation was performed in a normal phase with an ion exclusion column (Supelcogel C610-H) (6% Crosslinked, 9 μm particle size, L × I.D. 30 cm × 7.8 mm, Sigma-Aldrich, Saint-Louis, MO, USA). The temperature was set at 35 °C and the flow rate was set at 0.5 mL min^−1^. The solvent was H_3_PO_4_ 0.1%, and the separation was performed in the isocratic mode for 35 min. Standard solutions were prepared with chemicals from Roth.

#### 4.3.5. Statistics

Results are presented as mean values from three culture replicates. *t*-tests were performed to determine significant differences (*p* < 0.05) between two mean values.

## 5. Conclusions

We have used batch growth experiments to compare the potential of CO_2_-sufficient (light-limited) photoautotrophy and heterotrophy for the growth and pigment content of four green microalgae. Growth curve analysis showed that two of the four species (*C. sorokiniana* and *S. vacuolatus*) show heterotrophic growth rates that are similar to photoautotrophic growth rates in exponential phases, even when high light intensity is used for photoautotrophy. This holds true for both glucose and acetate as substrates for heterotrophy. On glucose, the longer duration of the exponential phase in heterotrophy, compared to photoautotrophy, allows heterotrophic cultures to reach much higher volumetric biomass productivities in heterotrophic conditions. However, this advantage of heterotrophy is counterbalanced by the lower biomass content values in carotenoids and chlorophylls. This can be partly alleviated by applying weak light during heterotrophic growth. Pigment content values are influenced by the carbon source used in heterotrophy. In *C. sorokiniana* and *S. vacuolatus*, pigment content values were higher on acetate, but this effect was not found in *S. acutus*. *C. reinhardtii* maintained high levels of carotene in acetate-supported heterotrophy. When considering pigment changes during photoautotrophic growth, we showed pronounced differences in pigment content between the exponential and deceleration (linear) phases due to the acclimation to lower perceived light during batch growth.

## Figures and Tables

**Figure 1 plants-13-01182-f001:**
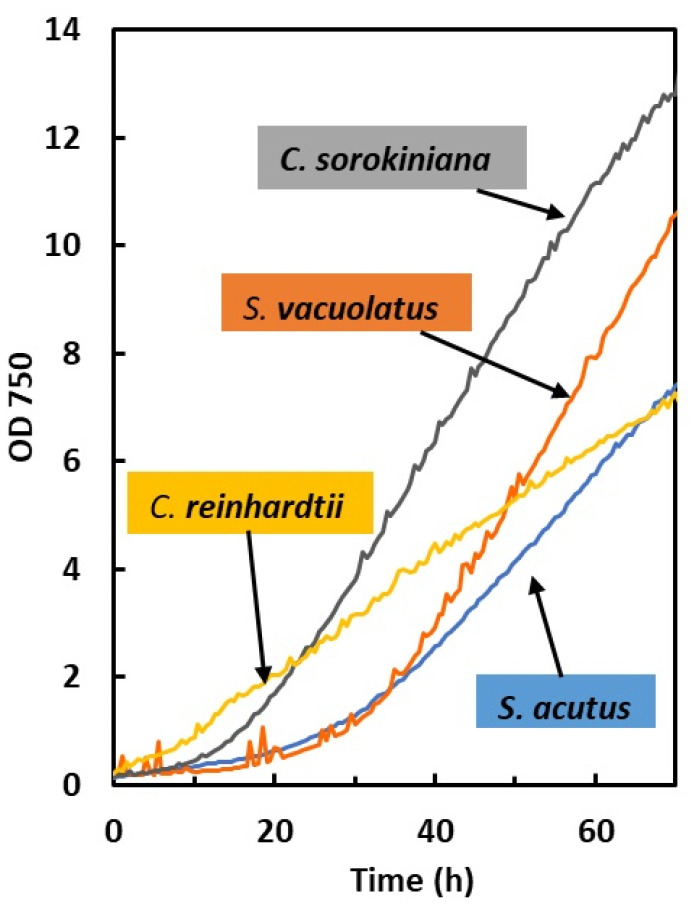
Typical photoautotrophic OD growth curves obtained under 300 µmol m^−2^ s^−1^ in light-limited conditions (bubbling with 5% CO_2_ in air). Note that the relationship between OD and biomass concentration is different depending on species. Biomass concentrations after 72 h were 2.02, 1.86, 2.21, and 1.77 g L^−1^ for *S. acutus*, *S. vacuolatus*, *C. sorokiniana*, and *C. reinhardtii*, respectively.

**Figure 2 plants-13-01182-f002:**
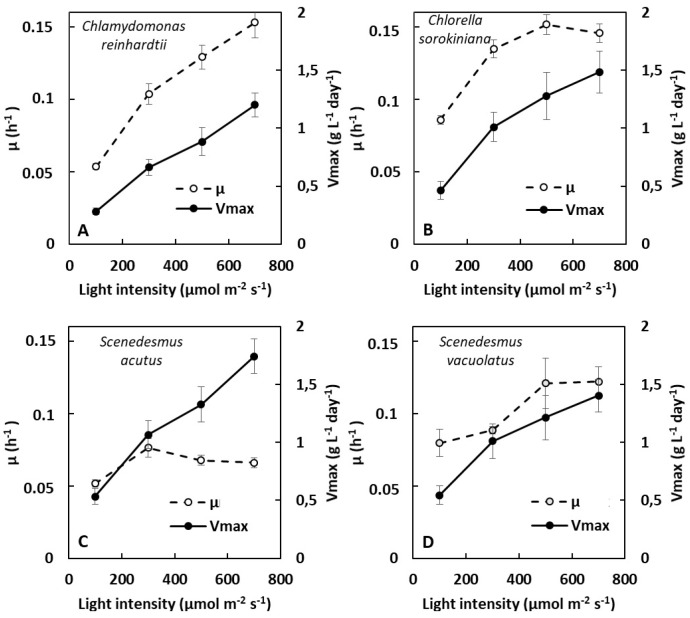
Light-limited photoautotrophic growth of four green microalgae: *C. reinhardtii (***A**), *C. sorokiniana* (**B**), *S. acutus* (**C**), and *S. vacuolatus* (**D**). The effect of light intensity on specific growth rate (µ) and on maximum volumetric biomass productivity (V_max_) (n = 3). µ (h^−1^) was measured during the exponential phase and on V_max_ (g L^−1^ day^−1^) during the deceleration phase.

**Figure 3 plants-13-01182-f003:**
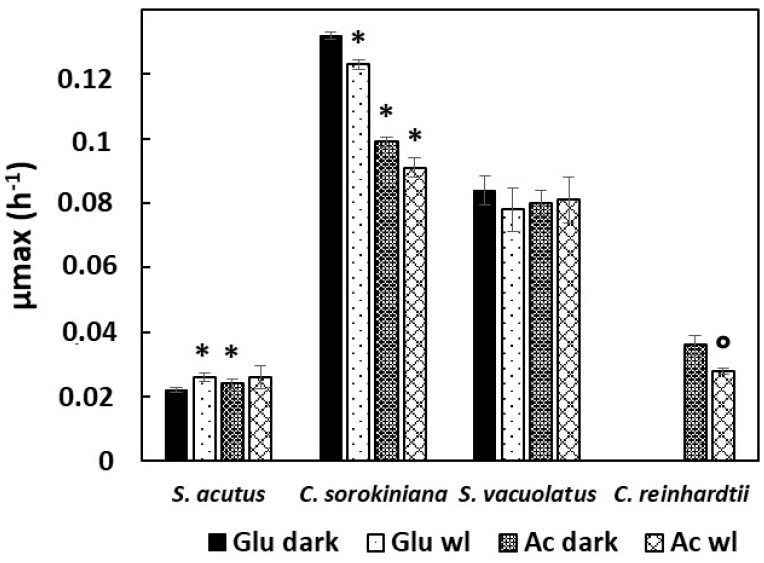
Heterotrophic growth rates of *S. acutus*, *C. sorokiniana*, *S. vacuolatus*, and *C. reinhardtii* using glucose (15 g L^−1^) or acetate (2 g L^−1^) as the carbon substrate (n = 3). µ_max_ values were measured during exponential growth either in complete darkness, or under continuous weak light (5 µmol m^−2^ s^−1^). * indicates significantly different values with reference to the value obtained in darkness using glucose as the substrate for a given species (*p* < 0.05) (°: for *C. reinhardtii* the reference is the value obtained in darkness using acetate as the substrate). Glu: glucose; Ac: acetate; wl: weak light.

**Figure 4 plants-13-01182-f004:**
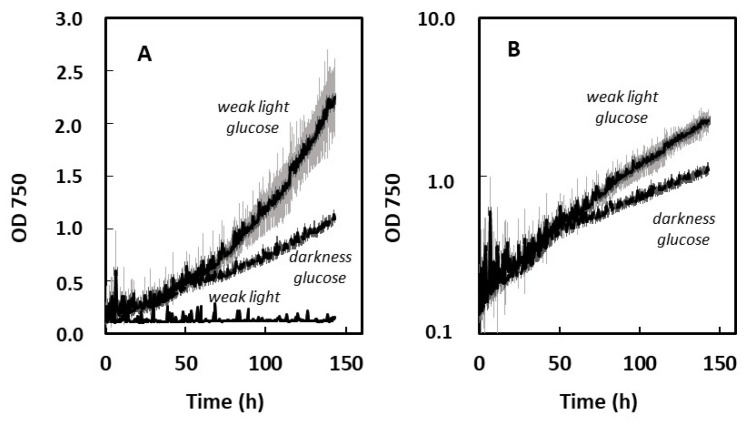
Stimulation of growth by weak light (5 µmol m^−2^ s^−1^) in *S. acutus* grown heterotrophically on glucose. Grey shading represents standard error (n = 3). (**A**) Growth curves. (**B**) Semi-logarithmic plots of growth curves. Biomass concentrations after 143 h were 0.41, 0.19, and 0.02 g L^−1^ for glucose + weak light, glucose in darkness, and weak light only, respectively.

**Figure 5 plants-13-01182-f005:**
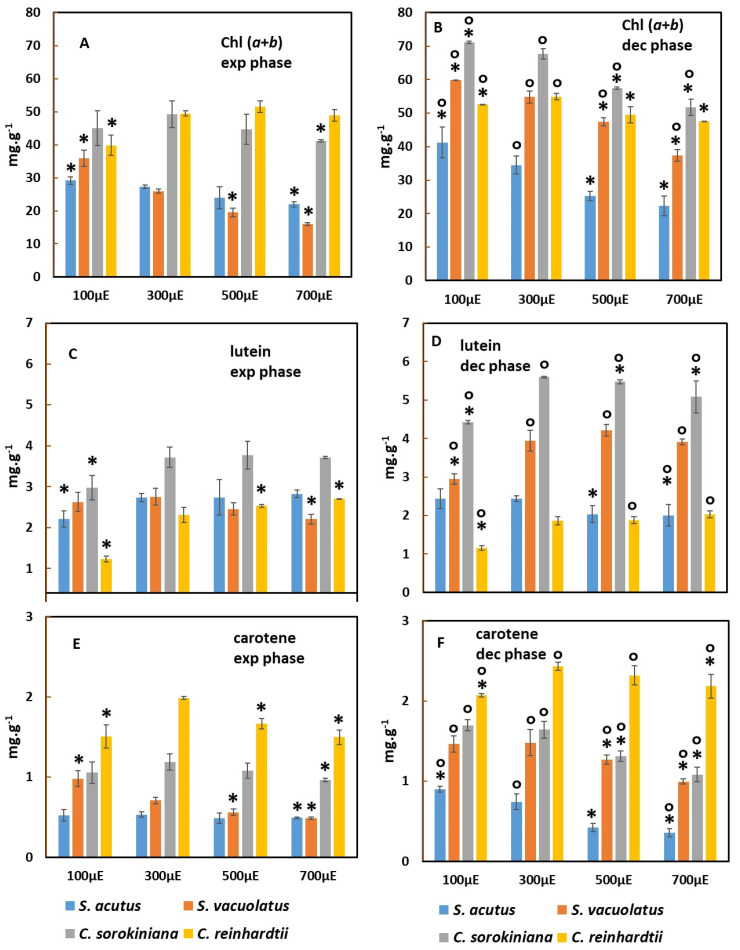
Effect of light intensity on biomass content in Chl (*a* + *b*) (**A**,**B**), lutein (**C**,**D**), and carotene (**E**,**F**) in *S. acutus*, *S. vacuolatus*, *C. sorokiniana*, and *C. reinhardtii* (n = 3). Pigment content is shown for the exponential phase (‘exp phase’: (**A**,**C**,**E**)) and for the deceleration phase (‘dec phase’: (**B**,**D**,**F**)) of light-limited photoautotrophic batch growth. * indicates for each species the significant effects of light intensity with values obtained at 300 µmol m^−2^ s^−1^ taken as reference (*p* < 0.05). ° indicates a significant phase effect with exponential phase values taken as a reference for each species (*p* < 0.05).

**Figure 6 plants-13-01182-f006:**
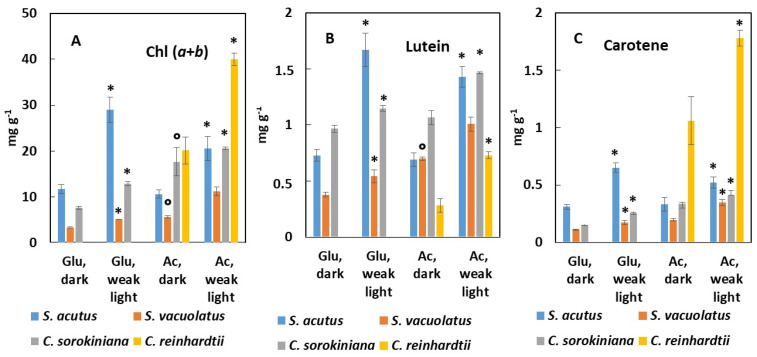
Chl (*a* + *b*) (**A**), lutein (**B**), and carotene (**C**) content of *S. acutus, S. vacuolatus, C. sorokiniana*, *and C.* grown heterotrophically on glucose or acetate (n = 3). Pigment content in the biomass is shown after 3 to 4 generations of batch growth in complete darkness or under weak light (5 µmol m^−2^ s^−1^). * indicates significant weak light effects (*p* < 0.05). ° indicates significantly different values on acetate in darkness with values on glucose taken as references (*p* < 0.05).

**Figure 7 plants-13-01182-f007:**
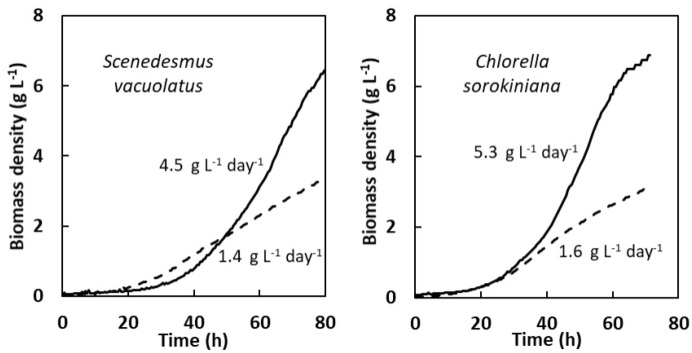
Comparison of heterotrophic (glucose 15 g L^−1^, full lines) and photoautotrophic (700 µmol m^−2^ s^−1^, dotted lines) biomass density growth curves for two fast growing strains *S. vacuolatus* and *C. sorokiniana*. Maximal biomass volumetric productivities are indicated.

**Figure 8 plants-13-01182-f008:**
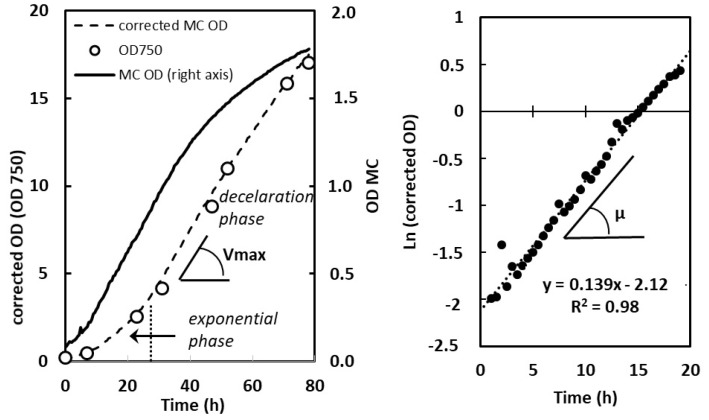
Correction of the OD curves obtained with an MC-100 multicultivator and determination of µ and V_max_ on the basis of OD corrected growth curves. Example with *C. sorokiniana* grown photoautotrophically at 500 µmol m^−2^ s^−1^ under bubbling with air enriched with 5% CO_2_ (1 point every 30 min). **Left**: Continuous line (right *Y*-axis): OD(720 nm) curve obtained from the multicultivator monitoring system. Circles (left *Y*-axis): true OD values measured in the course of the culture at 750 nm with a conventional spectrophotometer. Dotted line (left *Y*-axis): corrected growth curve using the OD(720 nm) values from multicultivator, corrected according to the following equation: $\frac{OD(750)}{OD(MC)}=a*e^{b*OD(MC)}$ with *a* = 1.795 and *b* = 0.934. V_max_ determination (V_max_ is measured as OD unit·L^−1^·day^−1^ and is further converted to g L^−1^ day^−1^). **Right**: µ determination from the semi-logarithmic plot of the exponential part of the corrected growth curve.

**Table 1 plants-13-01182-t001:** Biomass yields on substrate (Y_X/S_ in g(DW) g^−1^, n = 3) obtained on glucose or acetate during heterotrophic growth of *Chlorella sorokiniana*, *Scenedesmus vacuolatus*, and *Chlamydomonas reinhardtii* (Y_X/S_ was not determined for *Scenedesmus acutus*, due to its slow heterotrophic growth).

Species	Y_X/glucose_	Y_X/acetate_
*Chlorella sorokiniana*	0.55 ± 0.02	0.31 ± 0.03
*Scenedesmus vacuolatus*	0.51 ± 0.01	0.32 ± 0.01
*Chlamydomonas reinhardtii*	-	0.32 ± 0.01

**Table 2 plants-13-01182-t002:** Comparisons of pigment content values in heterotrophy, expressed as a percentage of the average content during the deceleration phase of light-limited photoautotrophy (based on data in Figure 5 and Figure 6). Values in darkness (‘dark’) and under weak light (‘w.l.’).

Species	Substrate	Carotene		Lutein		Chl (a + b)	
		dark	w.l.	dark	w.l.	dark	w.l.
*C. sorokiniana*	Acetate	23	28	21	29	28	33
	Glucose	10	17	19	23	12	21
*S. vacuolatus*	Acetate	14	26	18	26	11	22
	Glucose	08	12	10	14	07	10
*S. acutus*	Acetate	56	90	31	65	35	68
	Glucose	53	112	33	76	39	97
*C. reinhardtii*	Acetate	47	78	16	41	40	80

## Data Availability

Data are contained within the article and Appendix A.

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
