# Peer review of "Heterotrophy Compared to Photoautotrophy for Growth Characteristics and Pigment Compositions in Batch Cultures of Four Green Microalgae"

_plants, 2024, doi:10.3390/plants13091182_

Round 1
Reviewer 1 Report (Previous Reviewer 3)
Comments and Suggestions for Authors
In this study, the authors reported “Heterotrophy Compared to Photoautotrophy for Growth
Characteristics and Pigment Compositions in Batch Cultures of Four Green Microalgae”. Four strains of green microalgae were compared for growth and pigment composition under photoautotrophic or heterotrophic conditions. In this study, authors obtained some positive results. However, authors need a few improvement before published in this journal.
1. The device used in this study should provide the information of city and country, for example the HPLC device (Shimadzu, city, country), Tissue 709 Lyser II disrupter (Qiagen), MC 1000- 629
OD-8X (Photon Systems Instruments).
2. The medium of 3N-BBM, TAP, TGP should provide the references.
3. In 4.3.1, 50 ml should be 50 mL. In 4.3.4, there were two spaces in the first line.
Author Response
please see the attachment

Reviewer 2 Report (New Reviewer)
Comments and Suggestions for Authors
I think this is an OK paper, though it could still benefit from additional edition. It may not be ground breaking, but it does provide at set of data also others could see as helpful. Particularly, the Results section and to some extend the Discussion section are, however, not written too elegantly. The text could really benefit from additional revision by an experienced author. I don't know if this is the journal style, but I think the paper would have been easier to read if M and M had been presented before the results. This would have allowed you to focus the Results section more on the results with no needs for small comments on how you did things. My recommendations are minor revision.
Comments
p. 2, line 60. Glucose has also been used in 10x higher concentrations, at least in microalgal fed batch cultures?
p. 3, line 103-105. It would be more accurate to write e.g. 'avoid CO2 limited growth' instead of 'maximal biomass productivity', which is not a well-defined entity.
p. 4, Figure 1. Text box with biomass concentrations should, in my mind, be omitted and the information be given in the legend instead.
p. 4, line 121. It is not clear to me what you mean by ‘…is usually characterized by the specific growth rate (μ in h-1)’. The specific growth rate is constant in this phase.
p. 4, line 122. It is not quite clear to me, if what you call the ‘deceleration phase’ is the same as most others (I think) call the linear growth phase? It it true that the specific growth rate is decelerating in this phase, while the growth rate is constant.
p. 6, line 146. You would probably not expect saturation because of self-shading but rather a linear relationship with incident light intensity?
p. 6, line 177. You have not yet described what you have corrected?
p. 7, line 211. why do you think weak light would stimulate growth? It will not provide energy in substantial quantities. If you don't back up your experiments with theory, it is hard for the reader to understand, why this was important to investigate.
p. 9, line 274-279 and elsewhere. This is an example where you, in my mind, mix introduction into the Results section, and this makes reading a bit tedious.
p. 9, line 280. ‘Photoacclimated to a light intensity’ in batch cultures may lack behind cell growth since the cells need time to adapt why self-shading continuously increase with cell density, see e.g. Slot et al. (2006) Enzyme Microb Technol 38: 168-175.
p. 14, line 392-402. I will recommend that you avoid discussing issues, which you have not investigated in this study. In this case, photobioreactor design seems irrelevant to discuss since you have nothing new to bring forward. If you think it is important, it would be more appropriate to bring this information in the introduction.
p. 14, line 416-429. Why would you think there would be such a relationship? And, where do you show that such relation does not exist? Please provide the theoretical background as well as the data. Else, I suggest you delete Line 416-429.
p. 14, line 436. How come are fed batch cultures relevant? Such cultures have not been investigated in your study?
p. 15, Figure 7. I don’t think it is a good idea to show new results in the Discussion section. Results are best shown in the Results section.
p. 15, line 459-460. I don't believe you can conclude much about ‘heterotrophic growth rate for a given species seems only weakly dependent on the nature of carbon substrate’ based on such a small sample size of experiments.
Comments on the Quality of English LanguageI have the feeling that the paper is written by relatively inexperinced authors and could be improved by additional revisions
Author Response
please see the attachment

Reviewer 3 Report (New Reviewer)
Comments and Suggestions for Authors
I think this paper is quite interesting and have a moderate to high level of interest, particularly regarding the comparison of heterotrophy vs photoautrophy, particularly the addition of weak light over the heterotrophic studies to observe how that could beneficiate growth and pigment composition, for 4 different species of green microalgae - Scenedesmus acutus, Scenedesmus vacuolatus, Chlorella 18 sorokiniana and Chlamydomonas reinhardtii.
The objectives of the authors were clearly to show if there were differences between heterotrophy and photoautotrophy regarding growth and pigment composition of 4 different green microalgae, and if there were differences regarding carbon sources – acetate and glucose - (adding to weak light). The methodology is well done, and I particularly highlight the controls and fact that experiments had 3-4 generations considering the acclimatization process. The results were quite interesting, but I stood out that could be presented in a different way (particularly table 2) and captions should have more information regarding the presented results. I liked your discussion, because you explain how weak light can in certain way, with darkness conditions, influence the genes and metabolism to produce more quantity of pigments of interest.
Please see my comments attached in the manuscript.

The english seems clear to me. There are only minor typos.
Author Response
please see the attachment

Reviewer 4 Report (New Reviewer)
Comments and Suggestions for Authors
The paper is based on measurements that are not giving the possibility of a real comparison between the growth mechanisms which are already hard to compare. The structure of the paper is atypical and there are some fundamentally wrong concepts discussed. For example, in Figure 2, the authors are calculating the maximum specific growth rate for 4 light intensities, when it should be only one maximum for every light intensity. In the kinetic modeling of the growth process the maximum is a constant, not a function of light.
There are dry mass values for the photosynthetic growth, but there is no dry mass value for the heterotrophic growth. If we compare the OD, in figure 1 it reaches 12 and in figure 4 it reaches 1 (in the same time span). Can we say that in heterotrophy the microalgae are 12 times lower in concentration? Is the measuring comparable?
Since the title announce a comparison, the conclusions shall summarise the finding in this spirit.
Round 2
Reviewer 4 Report (New Reviewer)
Comments and Suggestions for Authors
I read the entire manuscript and it is much improved compared to the first version. The authors considered most of the comments from the reviewers. I have no additional comments, the article can be published in this form.
This manuscript is a resubmission of an earlier submission. The following is a list of the peer review reports and author responses from that submission.
Round 1
Reviewer 1 Report
Comments and Suggestions for Authors
1. It is recommended to discuss the mechanism of Pigment content of heterotrophically-grown cells: effect of carbon source (glucose or acetate) and of weak light to improve the quality of this article.
2. It is recommended to change the black/white pictures into color pictures to make it easier for readers to read.
3. The text in some pictures is not clear, it is recommended to redraw it.
4. The authors can conduct a more in-depth analysis of Figure 5.
5.
It is recommended to cite the article “CO2 favors the lipid and biodiesel production of microalgal-bacterial granular sludge” to improve the argument of this manuscript.
6. Page 7,“DNA extractions were performed following a modified protocol from Newman et al. (Newman et al. 1990).” Need to be modified to use the same citation format.
Comments on the Quality of English Language
no
Reviewer 2 Report
Comments and Suggestions for Authors
Dear editor, dear authors,
Here is my review of the article Plants-2664038 entitled "Heterotrophy Compared to Photoautotrophy for Growth Characteristics and Pigment Compositions in Batch Cultures of Four Green Microalgae".
The manuscript compare the growth and pigment composition of for microalgal species cultivated in heterotrophic, mixotrophic and photoautotrophic conditions. As heterotrophic source of carbon they used glucose and acetate. The authors measured growth of the cultures by biomass dry weights and specific growth rate. Pigments were quantified by HPLC analysis, Acetate was also determined by HPLC analysis (Why?).
General recommendations:
The research topic is suitable for publication, but the manuscript has some shortcomings related to writing of results and discussion. The article is very difficult to read and not well organized. In the introduction, the purpose of the research is somewhat clear, but there are no clear objectives of what the authors wanted to investigate and find out.
1. It needs more clearly and better organized results without introduction, objectives, discussion, all with references. Results have to be presented clearly without references to other results, without references about methods,without objectives with references . All discussion parts and material reference end explanation in results must be omitted from it and relocated them to other places.
2. There is no statistic calculations of results.
3. Subtitles of subsections inside results have to be more clearly written and can inform the reader about the content of it. Subsections have to follow the subchapters inside Sampling and analysis.
After my opinion, after improvement tthe manuscript can be resubmitted again !
Kind regards, Reviewer
Reviewer 3 Report
Comments and Suggestions for Authors
In this study, the authors reported “Heterotrophy Compared to Photoautotrophy for Growth Characteristics and Pigment Compositions in Batch Cultures of Four Green Microalgae”. Four strains of green microalgae were compared for growth and pigment composition under photoautotrophic or heterotrophic conditions. In this study, authors obtained some positive results. However, authors need a few improvement before published in this journal.
1. In fig 1, author did not show the completed growth curves so that I can not find the time point of highest cell density. And please show different growth curves by different symbols. I am curious that why choose deceleration phase to sample? Why did you choose the phase that the cell density is the highest?
2. You can use abbreviation when the species name appear for the second time. For example, Chlamydomonas reinhardtii would be C. reinhardtii for the second time. Please check in the whole article.
3. Please use different letters to distinguish different results in one figure such as fig2 and fig 4.
4. Please give a clear description to fig 4 because there were two figures and I am confused.
5. The results in fig 5 and 6 were lack of significant difference analysis. Please add them.
6. Please change ml to mL in the whole article.
7. Did the four microalgae use the same medium when cultured in multi-cultivators? How to maintain the pH? Did you add supplementary culture medium in the cultivation? How about the air flow in heterotrophic cultures? Authors should make it clear in introduction of Cultivation conditions.
Comments on the Quality of English LanguageIt is OK.